# *Staphylococcus aureus* and MRSA in Livestock: Antimicrobial Resistance and Genetic Lineages

**DOI:** 10.3390/microorganisms11010124

**Published:** 2023-01-03

**Authors:** Vanessa Silva, Sara Araújo, Andreia Monteiro, José Eira, José Eduardo Pereira, Luís Maltez, Gilberto Igrejas, Teresa Semedo Lemsaddek, Patricia Poeta

**Affiliations:** 1Microbiology and Antibiotic Resistance Team (MicroART), Department of Veterinary Sciences, University of Trás-os-Montes and Alto Douro (UTAD), 5000-801 Vila Real, Portugal; 2Department of Genetics and Biotechnology, University of Trás-os-Montes and Alto Douro (UTAD), 5000-801 Vila Real, Portugal; 3Functional Genomics and Proteomics Unit, University of Trás-os-Montes and Alto Douro (UTAD), 5000-801 Vila Real, Portugal; 4Associated Laboratory for Green Chemistry (LAQV-REQUIMTE), University NOVA of Lisboa, 1099-085 Lisbon, Portugal; 5Veterinary and Animal Research Centre (CECAV), University of Trás-os-Montes and Alto Douro (UTAD), 5000-801 Vila Real, Portugal; 6Associate Laboratory for Animal and Veterinary Sciences (AL4AnimalS), University of Trás-os-Montes and Alto Douro (UTAD), 5000-801 Vila Real, Portugal; 7Centro de Investigação Interdisciplinar em Sanidade Animal (CIISA), Faculdade de Medicina Veterinária, Av. Universidade Técnica de Lisboa, 1300-477 Lisboa, Portugal

**Keywords:** *Staphylococcus aureus*, MRSA, LA-MRSA, livestock, genetic lineages

## Abstract

Animal production is associated with the frequent use of antimicrobial agents for growth promotion and for the prevention, treatment, and control of animal diseases, thus maintaining animal health and productivity. *Staphylococcus aureus*, in particular methicillin-resistant *S. aureus* (MRSA), can cause a variety of infections from superficial skin and soft tissue infections to life-threatening septicaemia. *S. aureus* represents a serious public health problem in hospital and community settings, as well as an economic and animal welfare problem. Livestock-associated MRSA (LA-MRSA) was first described associated with the sequence (ST) 398 that was grouped within the clonal complex (CC) 398. Initially, LA-MRSA strains were restricted to CC398, but over the years it has become clear that its diversity is much greater and that it is constantly changing, a trend increasingly associated with multidrug resistance. Therefore, in this review, we aimed to describe the main clonal lineages associated with different production animals, such as swine, cattle, rabbits, and poultry, as well as verify the multidrug resistance associated with each animal species and clonal lineage. Overall, *S. aureus* ST398 still remains the most common clone among livestock and was reported in rabbits, goats, cattle, pigs, and birds, often together with *spa*-type t011. Nevertheless, a wide diversity of clonal lineages was reported worldwide in livestock.

## 1. Staphylococci—An Overview

Staphylococci are Gram-positive cocci-shaped cells arranged in irregular clusters similar to grape clusters [1]. Although *S. aureus* is the major bacterium of its genus, more than 50 Staphylococcus species are registered in the *List of Prokaryotic Names with Standing in Nomenclature database* [2]. These bacteria are facultative anaerobes, catalase positive and salt (NaCl) tolerant [3]. The Staphylococcus genus is divided in two main groups, the coagulase-negative staphylococci (CoNS) and coagulase-positive staphylococci (CoPS). *Staphylococcus* species were first characterized by Friedrich Rosenbach, who established that yellow/orange colonies corresponded to CoPS species and white colonies to CoNS [4]. In general, staphylococci are natural inhabitants of skin and mucous membranes of human beings and animals, while the prevalence of species varies widely according to the host [2]. However, they can cause various types of infections ranging from skin infections to severe infections like necrotizing pneumonia or sepsis [5]. Staphylococci tolerate a wide range of temperatures, from 7 to 40 ℃, and survive under unfavourable conditions like dryness and dehydration, making them a persistent and widespread pathogen in the environment [3]. Among Staphylococci, *Staphylococcus aureus* and *Staphylococcus epidermidis* are the most frequently isolated species of the CoPS and CoNS groups, respectively, which is why *S. epidermidis* is the species most focused on in CoNS studies [4]. *Staphylococcus aureus* has been the major pathogen associated with nosocomial infections in humans [1]. However, more recently, infections caused by CoNS like *S. epidermidis, S. haemolyticus* and *S. saprophyticus* have emerged in hospitals [6]. Moreover, although to a lesser extent, CoNS are known to colonize both farm and domestic animals. For instance, several bovine mastitis infections associated with CoNS have increasingly been reported over the years [7,8,9]. CoNS have also been detected as contaminants in food products [10,11,12,13] becoming more frequent and harmful to both animals and humans, affecting public health and taking a huge economic burden.

## 2. Methicillin-Resistant Staphylococci

Penicillin is an antibiotic in the β-lactam family. It was introduced in clinical practice in 1941 and, only a year later, penicillin resistance in *S. aureus* was described. Ten years later, more than half of the isolates in large hospitals were already resistant to penicillin [14]. The need to create compounds resistant to the action of β-lactamases produced by staphylococci led to the development of other antimicrobials such as methicillin. However, shortly after the introduction of this antibiotic, methicillin-resistant *S. aureus* (MRSA) was isolated [15]. MRSA is now recognized as a high-priority pathogen according to the World Health Organization (WHO), since an emergence of these strains has been observed [16]. The mechanism of resistance to methicillin is based on the synthesis of a new penicillin-binding protein (PBP), the PBP2a, with little affinity for methicillin and other β-lactams, which blocks the arrival of the antibiotic at its target site and thus creates resistance [17]. The *mec*A gene is the main gene responsible for resistance to methicillin. This gene is found on the genetic loci called staphylococcal cassette chromosome *mec* (SCC*mec*). The expression of the *mec*A gene depends on two genes, *mec*R1, which regulates transcription, and *mec*I, which encodes the repressor protein [18]. There are two other mechanisms that result in weak resistance to methicillin and oxacillin in which the presence of the *mec*A gene is not clear. Strains with modifications in the affinity of PBP 1, 3, and 4 exhibit weak resistance to methicillin, and strains that hyper-produce β-lactamases have a limited resistance to oxacillin [19]. Although the *mec*A gene is the most common PBP2a-encoding gene, recently, the detection of a new *mec* gene was reported. *mec*C is part of the SCC*mec* type XI and has been detected in both *S. aureus* and CoNS in animal and environmental samples [20,21]. Two less frequent *mec* genes have been described, *mec*B and *mec*D, both detected in *Macrococcus caseolyticus*, although the *mec*B gene has been recently reported in *S. aureus* [22,23].

*Staphylococcus* spp. have the ability to quickly acquire resistance to antibiotics. This started with the first strains resistant to penicillin reported a few years after its introduction. The successive introduction of other antibiotics, such as macrolides, tetracycline, and chloramphenicol, had a similar result: that is, the rapid appearance of resistant strains which, in turn, led to the proliferation and dissemination of bacteria with a broad spectrum of resistance and a remarkable ability to survive in different environments [24,25]. One interesting fact about most of these strains is that they are generally resistant to β-lactams [26]. Different species have a wide range of antibiotic resistances and monoresistant and multiresistant strains have been found in the environment, in particular in the aquatic environment [27].

### 2.1. Mechanisms of Antibiotic Resistance in S. aureus

Antimicrobials are substances that inhibit or eliminate the growth of microorganisms and are therefore commonly used in the treatment and prevention of infections. They can be classified as bactericidal or bacteriostatic. Bactericidal drugs act by directly eliminating the microorganism, while bacteriostatic drugs inhibit bacterial growth and multiplication [28]. Among the main groups of bactericidal antibiotics are aminoglycosides, β-lactams, fluoroquinolones, and glycopeptides, while lincosamides, oxazolidinones, and macrolides are some of the drugs with bacteriostatic action [29].

Bacterial resistance to antimicrobials can be manifested through several mechanisms such as inhibition of cell wall synthesis, inhibition of cytoplasmic membrane synthesis or damage, inhibition of protein synthesis in ribosomes, changes in the synthesis of nucleic acids, and alteration of cellular metabolisms [30]. The resistance of bacteria to antibiotics can have an intrinsic typology, depending a lot on the typical and specific biology of the bacterium, or extrinsic, through horizontal gene transfer, where there is the acquisition of new genes, transported by plasmids, transposons, integrons, or bacteriophages, also known as phages [31].

Genetic alterations promoted by the mechanisms described here may provide bacteria with the ability to carry out certain biochemical mechanisms such as: enzymatic inactivation, efflux pumps, or target modification which can be specific for different classes of antimicrobials, as shown in Table 1 [32,33].

### 2.2. Virulence Factors in S. aureus

*S. aureus* is one of the first pathogens described and remains one of the most common causes of life-threatening infections of the bloodstream, skin, soft tissues, and even the lower respiratory tract, and serious deep-seated infections such as endocarditis, mastitis, and osteomyelitis [34]. The degree of pathogenicity of *S. aureus* depends on several components of the bacterial surface and extracellular proteins. The expression of most virulence factors in *S. aureus* is controlled by the *agr* locus (accessory gene regulator) which encodes a two-component signaling pathway whose activation ligand is a bacterial density sensor peptide (autoinducer peptide) which is also encoded by *agr* [35]. Thus, the agr system is directly involved in controlling the expression of virulence factors in *S. aureus* [35].

Four hemolysins have been identified: alpha, beta, gamma, and delta. They have hemolytic and cytolytic capacity and act on some cells of the host such as macrophages, leukocytes and fibroblasts. α-hemolysin is the most studied. It is responsible for the hemolysis zone observed around the *S. aureus* colonies and has its effect on eukaryotic cell membranes and erythrocytes. β-hemolysin is an active sphingomyelinase that acts on different cells, and γ and δ hemolysins, found in some strains of *S. aureus*, have the ability to lyse a variety of different cells [36]. Only 5–10% of *S. aureus* produce exfoliative toxins. Two serotypes of exfoliative toxins have been reported, A and B, which are biochemically and immunologically different, but with similar biological functions [37]. Exfoliative toxin A is chromosomal encoded whereas B is plasmid encoded. Both have proteolytic activity, dissolve the mucopolysaccharide matrix of the epidermis, and act as superantigens [38]. Enterotoxins are responsible for food poisoning with emesis and enterocolitis symptoms and are produced by 30–50% of *S. aureus* strains [39]. Toxic shock toxin is a thermostable protein synthesized by chromosomal genes. They are considered superantigens since they are capable of producing the massive proliferation of T lymphocytes and thus the production of cytokines, which leads to toxic shock syndrome [38]. Panton–Valentine’s leucocidin is a cytolytic toxin and is formed by two proteins, LukS-PV and LukF-PV, which form pores across the membrane of leukocytes causing efflux of the cell content, leading to lysis [40].

## 3. The Issue of Antimicrobial Resistance in Livestock

In animal production farms, the main goal is to raise healthy animals for food consumption by humans. In such places, animals should be provided with good nutrition, a clean and dry environment, good ventilation, and supervision by veterinarians [41]. Just like in humans, animals get sick with bacterial infections and need antibiotic treatment. Treatment of sick animals with antibiotics is of prime importance because sometimes there are no other alternatives, and that is the only choice to stop the illness spreading to the rest of the herd or flock, promoting animal welfare, and contributing to food safety. However, antibiotics have been used for a long time for non-medical purposes. In subtherapeutic levels, these medicines improve growth rate, reduce mortality and morbidity, and improve reproductive performance in production animals, allowing industries to achieve a highly efficient production [41]. Besides all the benefits provided by antibiotics with respect to animal health and welfare, the continuous use of those medications at low levels brings risks to public health due to the emergence of resistant microorganisms.

In low-middle-income countries, antimicrobials are frequently used as prophylactic drugs in animal production. Because of the easy access to these medicines and their non-restricted use, antimicrobial consumption in the animal production system almost double human consumption [42]. In developing countries unqualified animal healthcare providers play a significant role in this problematic situation [42].

The use of antibiotics in food animals for purposes of disease treatment and prevention and growth promotion are frequently associated with the global rise of AR. In fact, estimated global antibiotic consumption in 2010 was 63,151 ± 1, 560 tons and it is estimated that, by 2030, this number will increase by 67% [43,44].

In order to promote responsible use of antibiotics, international and national guidelines are available with the purpose of ensuring therapeutic efficacy and mitigating antimicrobial resistance. However, compliance with such guidelines is very variable among countries.

Most of the literature focuses on the risks of bacteria resistance directly transferred from production animals to people, affecting public health. However, the consequences of antibiotic resistance of animal origin are beyond that. In production animals, the bacterial diseases can bring on, behind all the suffering, huge economic losses. In several species, respiratory and enteric diseases are very common, while in others—like cows, goats and, sheep and other animals kept for milk production—mastitis is the major problem [45]. Because these infections are contagious, the problem grows when animals are kept in large groups and close to each other. In aquaculture, bacterial diseases are also important because the animals, such as fish and shrimps, are raised in large numbers with close contact [45].

The first cases of antibiotic resistance in food animals were observed in turkeys in 1951, after streptomycin was fed to those animals [46]. Since then, resistance to antibiotics, such as tetracyclines, sulfonamides, β-lactams, and penicillin, have increasingly been observed [43].

In terms of public health, two major routes of antimicrobial-resistant bacteria transmission to humans exist. They are direct acquisition through contact with food producing infected animals or indirect acquisition through the food chain or via exposure to niches with high antimicrobial pollution [47]. Several studies have found a high prevalence rate of antimicrobial resistance among individuals like farm workers and veterinarians that have a close relationship with food-producing animals [48,49,50].

Relative to environmental pollution caused by antibiotics in animal-producing sectors, it is known that only a small amount of the administered antibiotics participates in animal metabolism and are effectively used [51]. In fact, it is estimated that 30–90% of antibiotics are excreted in animal urine or faeces in the form of main metabolites [52]. High concentrations of these metabolites can enter the soil and water environment in various ways, causing pollution to the ecological environment [52]. Subsequently, the microbial species that occupy the soil or water will suffer a selective pressure that results in the emergence of antibiotic-resistant phenotypes. It is a very similar process that occurs when we talk about the antibiotic use in agriculture. If these phenotypes emerge on a mobile genetic element, like a plasmid, they can be horizontally transmitted to animals or humans, which may pose a considerable public health threat [47].

Taken together, it can be assumed that the use of antibiotics in animal settings, specifically their massive use for growth promotion, can lead to the emergence of antibiotic-resistant bacteria. That will affect not only animal wellbeing, but also public health, because humans interact directly or indirectly with them. Finally, the economic burden from antimicrobial resistance is still a problem worldwide.

## 4. *Staphylococcus aureus* in Livestock

In this review article, we gathered information from studies that detected and characterized *S. aureus* in pigs, goats, horses, sheep, buffaloes, cattle, rabbits, and poultry. LA-MRSA isolates are genetically distinct from human isolates. Table 2 presents a general summary of the molecular characteristics of LA-MRSA clones worldwide.

Although few, some studies investigated the presence of *S. aureus* in rabbits (Table 3). A study carried out by Ruiz-Ripa et al., in rabbits from Spain, in which nasal and rectal samples from 103 mammals were analysed, detected the presence of three *mec*C-positive MRSA isolated from rabbits, which belonged to the ST130 clonal lineage, the complex clonal CC130 and the *spa*-type t843; in addition, they showed resistance to cefoxitin and penicillin [53]. Another study also carried out in rabbits in China analysed 691 samples (lung, mastitis, and foot dermatitis) and isolated *S. aureus* strains belonging to clonal strains ST121 and ST398. In addition, most isolates had a multidrug-resistance phenotype showing resistance to many antimicrobials such as streptomycin, kanamycin, gentamicin, penicillin, ciprofloxacin, and azithromycin, even though the samples were collected from different provinces in China. Regarding virulence, *S. aureus* isolates had the genes encoding for the virulence factors *hla*, *hlb*, *clf*A, *clf*B, and *fnbp*A [54]. A less recent study, carried out by Holmes et al. in the United Kingdom, more specifically in England and Scotland, reported a wide variety of genetic lineages among S. aureus isolated from rabbits: ST30, CC30, ST3126, CC291, ST15, CC15, ST6, CC6, ST3120, ST121, CC425, CC121, ST3092, ST39, ST2257, and CC22. Regarding *spa*-types, the following were detected: t021, t1614, t2574, t5413, t13114, t645, t15410, t15409, and t1977 [55]. The *S. aureus* isolates from the same study were mostly susceptible to the antimicrobials tested, with some isolates showing resistance to penicillin, fusidic acid, tetracycline, and fluroquinolones encoded by the genes *bla*Z, *tet* (38), and *nor*A. On the other hand, a high diversity of virulence genes were found among the isolates: *fnb*A, *fnb*B, *sdr*D, *sdr*E, *efb*, *cna*, *sea*, *seb*, *sei*, *sen*, *seo*, *seg*, *sep*, *tst*, *sak,* and *scn* [55]. Indrawattana et al. carried out a study in rabbits in Thailand, in which 67 purulent samples were studied. From these samples, different *S. aureus* strains were isolated and were ascribed to ST4209, ST4212, and ST4213. Furthermore, resistances to penicillin, aminoglycosides, and lincosamides were detected in *S. aureus* isolates conferred by the *bla*Z, *mec*A, *aac*A-*aph*D, and *mrs* (A) genes [56]. Another study conducted in Portugal by Silva et al. investigated the presence of MRSA in 66 purulent samples from rabbits. Most isolates were ascribed to ST2855 and *spa* type t1190. Nevertheless, other genetic lineages were also identified including CC97, ST105, CC5, ST5, ST582, CC15, ST22, CC22, ST1, and CC1 and *spa* types t002, t2802, t1094, t084, t032, and t1491 [57]. In the same study article, the authors also reported the presence of strains showing a multidrug-resistant phenotype with resistance to cefoxitin, penicillin, tetracycline, erythromycin, clindamycin, gentamicin, tobramycin, ciprofloxacin, and fusidic acid associated with the *mec*A, *tet* (K), *tet* (L), *erm* (B), *msr* (A/B), *lin*B, *erm* (C), *vga*B, *vga*A, *aac* (6′)-Ie-*aph* (2′’)-Ia genes. Finally, in that study, the presence of the virulence genes *hlb*, *cna* and *et*A, as well as the immune evasion cluster (IEC) type B and C-related genes *scn*, *chp,* and *sak*, was also detected [57]. In a study carried out with rabbits from Spain and Portugal, 240 samples were analysed. These samples were recovered from mastitis (*n* = 86), skin abscesses (*n* = 33), pododermatitis (*n* = 31), dermatitis (*n* = 21), otitis (*n* = 13), metritis (*n* = 12), peritonitis (*n* = 2), conjunctivitis (*n* = 11), pneumonia (*n* = 9), rhinitis (*n* = 3), hepatitis (*n* = 3), osteomyelitis (*n* = 1), pericarditis (*n* = 1), and nasal samples (*n* = 14). Among these samples, the prevalence of MRSA varied between 4.8% and 100% and the isolates were ascribed to ST146, ST398, ST2855, and ST4774 [58]. Some MRSA isolates carried the *mec*C gene, which confers resistance to methicillin, and were also ascribed to SCC*mec* types IV, V, and III. Regarding resistance and virulence, MRSA isolates presented a wide variety of resistances, including resistance to vancomycin, and some isolates were categorized as IEC type E [58].

**Table 3 microorganisms-11-00124-t003:** Molecular characteristics of LA-MRSA clones in different continents [22].

Animal	Location	Clonal Lineages		Virulence		References
		MLST	*spa* Type	IEC Types	Outros Genes	
**Rabbits**	China	ST121, ST398			*nuc, hla, hlb, clfA, clfB, fnbp*A	[54]
	England	ST30, ST3126, ST15, ST6	t021, t1614, t2574, t5413		*fnbA, efb, cna, sea, seb, sei, sen, sea, sep, tsst, sak, scn, sdr*D*, sdr*E*, efb*	[55]
	Scotland	ST3120, ST121, ST3092, ST39, ST2257,	t13114, t645, t15410, t15409, t1977		*fnbA,* fnbB*, sdr*D*, sdr*E*, cna, sea, sec, sei, sen, sep*	[55]
	Thailand	ST4209, ST4212, ST4213				[56]
	Switzerland	CC5, CC7, CC8, CC15, CC96, CC101, CC121, ST890	t160, t179, t8456, t14871, t091, t681, t4770, t085, t745, t1190, t056, t741, t1773		*sea, sea*, *seb, sed, egc, sej, ser*	[59]
	Portugal	ST2855, ST105, ST5, ST582, ST22, ST1	t1190, t002, t2802, t2802, t1094, t084, t032, t1491	B, C	*hlb*, *can, eta, cna*	[57]
	Spain	ST146, ST398, ST2855, ST4774		E		[58]
	Portugal	ST2855				[58]
**Goat**	Korea	ST72, ST398	t324, t571, t324, t664		*luk*ED, *seg, sei, sem, sen, seo, seq*	[60]
	Greece		t1336			[61]
	Iran	ST97, ST522/2057, ST2079, CC5, CC522,	t267, t1534, t7305, t5428			[62]
**Sheep**	Greece		t3586			[61]
	Czech Republic	ST398	t011			[63]
	Iran	ST522/2057, ST130, CC522, CC130	t1534, t16958, t12270			[62]
**Cattle**	Greece		t3586			[61]
	Czech Republic	ST398, ST361, ST1	t011, t034, t2346, t1255, t315, t127			[63]
	Czech Republic	ST398	t011			[63]
	Iran	ST6, ST15, ST291, CC5, CC398	t304, t084, t937			[62]
	India	ST5220				[64]
**Buffalo**	Philippines	ST2454, ST9	t091, t800			[65]
**Pigs**	Italy	CC97, CC398	t1730, t899, t4795			[66]
	Northern Ireland	ST30, ST398	t749, t011		*luk*M, *luk*F-P38	[67]
	Italy	ST398, ST9, ST97, ST1	t011, t034, t899, t1451, t1456, t4474, t588, t1793, t4795, t1730, t9301, t127			[67]
	Czech Republic	ST398	t011, t034, t2123, t2346, t4661			[63]
	Latvia		t011, t1333, t808, t808, t400, t400, t1580, t1985, t2383			[68]
	Switzerland	ST752, ST398	t011, t034			[69]
	Italy	ST398	t899, t034, t011			[70]
	Denmark	CC398	t034, t011			[71]
	Spain	ST398	t011, t034, t2346, t108, t1197, t1456, t1451			[72]
	Poland	ST398, ST9, ST433, ST12, ST2950	t011, t034, t108, t4389, t318, t1580, t1928, t4387, t1255, t337, t1430, t8893, t5817, t021, t156, t005		*seg, sei*	[73]
	Finland	CC1, CC398	t127, t1381, t011, t034, t108, t2741			[74]
	Italy	ST398	t011, t108, t899, t034, t571, t1606, t5524, t1793, t10485, t2876			[75]
**Piglets**	Portugal	ST398, ST1	t011, t1451, t108, t1491		*hla, hlb, hld, hlg, hlgy*	[76]
**Chickens**	Belgium	ST398, ST239	t011, t037, t899			[77]
	Denmark	ST398, ST8, ST5, ST1	t011, t108, t008/, t347, t002, t034, t273			[78]
	Switzerland					[79]
	The Netherlands	ST398	t011, t034, t108, t899, t3015			[80]
	Italy	ST612, ST8, ST30	t1257, t018			[81]
	Egypt					[82]
	South Africa					[83]
**Chickens and turkeys**	Germany	ST398, ST9, ST5, ST1791	t011, t034, t899, t023, t6574		*luk*D*, luk*E, *hla, hlb, cna, fib*	[84]
	Switzerland		t011, t034			[85]
	The USA	ST5, ST6, ST8, ST15, ST406, ST544, ST699, ST398				[86]
**Chickens, turkeys and geese**	Poland	ST398, ST5	t002, t034, t214, t436, t127, t912, t663, t1422		*egc1*, *seh*	[87]
**Quails**	Portugal	ST398, ST6831	t011, t9747		*hla, hlb, hld, scn*	[88]
**Ducks**	The Netherlands	ST398	t011			[89]
**Turkeys**	Germany	ST398	t011, t002, t1456, t034			[90]

Abbreviations. MLST: multilocus sequence typing; IEC: immune invasion cluster; ST: sequence type.

Regarding samples from cattle, in a study carried out in Korea by Mechesso et al., nasal samples (*n* = 431) and carcass samples (*n* = 50) of goats were analysed. Among the samples, a prevalence of 1.2% of MRSA was detected and the strains belonged to ST72, ST398, *spa* types t324, t571, and t664 and SCC*mec* types IV and V. Finally, some virulence genes were identified including *luk*ED, *seg*, *sei*, *sem*, *sen*, *seo,* and *seq* [60]. Another study, carried out by Papadopoulos et al., in cows (n = 89), sheep (n = 104), and goats (n = 84), reported a prevalence of MRSA that varied between 1.10% and 2.90%. The isolates were ascribed to *spa* types t3586 and t1336 and showed resistance to penicillin, oxacillin, sulfamethoxazole-trimethoprim, gentamicin, kanamycin, and tetracycline [61]. In the Czech Republic, a study was carried out in cattle (n = 34), goats (n = 38), and sheep (n = 25), in which the prevalence of MRSA was between 0.6% and 11.30%, the isolates belonged mainly to ST398, ST361, and ST1 and the *spa* types t011, t034, t2346, t1255, and t315 [63]. A study conducted by Dastmalchi Saei et al., with goats (*n* = 10), cows *(n* = 9), and sheep (*n* = 8) samples from Iran detected *S. aureus* isolated belonging to ST6, ST15, ST291, CC5, CC398, ST522, ST2057, ST130, CC522, CC130, ST97, ST2079, CC522 and *spa* types t304, t084, t937, t1534, t16958, t12270, t267, t7305, and t5428. In addition, the isolates showed resistance to penicillin, erythromycin, clindamycin, and tetracycline [62]. Finally, another study conducted with cattle from Indian samples reported the detection of one *S. aureus* isolate which belonged to ST5220 and SCC*mec* V [64].

The presence of *S. aureus* and MRSA in the swine population was widely studied. In a study carried out in pigs in Italy, there was a prevalence of MRSA of 51.40% mostly associated with CC97, CC398 and *spa* types t1730, t899, and t4795. In addition, the MRSA strains showed resistance to cefoxitin, gentamicin, clindamycin, tetracycline, rifampicin, enrofloxacin, and erythromycin and carried the virulence genes *clf*A, *luk*E, *cna*, *luk*E-*luk*D, and *fmt*B [66]. In 2016, several studies associated with the detection and characterization of *S. aureus* in pigs were published. One of these studies was carried out in Northern Ireland, in which the clonal lineage ST30, the *spa* type t749 and SCC*mec* Vt were detected among *S. aureus* isolates. Regarding antimicrobial resistance, the strains showed resistance mainly to penicillin and tetracycline and carried the *bla*Z and tet (K) genes. On the other hand, with regard to virulence, the *luk*M and lukF-P38 genes were detected [67]. Another study carried out in 2016 in Ireland reported that all pig isolates belonged to CC398, *spa* types t034 and t011 and SCC*mec* Vt. In addition, some isolates were multidrug resistant and harboured the *bla*Z, *erm* (A), *erm* (B), *aac*A-*aph*D, *aad*D, *tet* (K), *tet* (M), *tet* (L), *fex*A, *dfr*G, and *dfr*K genes [91]. Another study conducted in Spain, more specifically in Catalonia, reported a prevalence of MRSA between 10% and 80% among pigs, with ST398 being the only sequence type detected. The isolates were ascribed to the *spa* types t011, t034, t2346, t108, t1197, t1456, and t1451. Finally, MRSA isolates were reported to be resistant to tetracycline, clindamycin, ciprofloxacin, erythromycin, gentamicin, tobramycin, and sulfamethoxazole-trimethoprim [72]. Another study carried out in Finland showed the presence of *S. aureus* CC1 and CC398, *spa* types t127, t1381, t011, t034, t2741, and t108, SCC*mec* V in pigs. These isolates also showed resistance to multiple antimicrobial classes conferred by the *mec*A, *bla* (Z), *tet* (K), *tet* (M), *aadD,* and *erm* (B) genes [74]. Other studies have been conducted in pigs with MRSA and *S. aureus* isolates being mostly associated with ST398 (CC398), and *spa* type t011, SCC*mec* type V and resistance to tetracycline [63,68,69,70,71,73,75]. A study conducted with piglets in Portugal revealed a prevalence of MRSA of between 98% and 100%. Once again, the clonal lineage ST398 and the *spa* types t011 and t108 were detected. Furthermore, all isolates were resistant to tetracycline and some were resistant to erythromycin, clindamycin, sulfamethoxazole-trimethoprim, and gentamicin [92]. In a study conducted in Poland, *S. aureus* isolated from pig samples were ascribed to a high diversity of genetic lineages including CC398, ST398, CC30, ST433, CC9, ST9, CC12, ST12, CC22, ST2950, *spa* types t011, t034, t108, t4389, t318, t1580, t1928, t4387, t1255, t337, t1430, t8893, t5817, t021, t156, and t005, and SCC*mec* types V (5C2&5), V (5C2), and IV (2B) [71]. In another study performed with pig samples from Italy, several different genetic lineages were also detected in MRSA isolates: ST398, ST9, ST97, ST1, the *spa* types t011, t034, t899 t1451, t1456, t4474, t588, t1793, t4795, t1730, t9301 and t127, and SCC*mec* V and IVa [93].

Regarding birds, a few studies conducted worldwide reported the presence of MRSA strains among food-producing birds including chickens, turkeys, geese, and ducks. MRSA CC398 was the most frequent clone detected in several countries across Europe in ducks, chickens, and turkeys (Table 3). Most strains from birds showed a multidrug-resistant profile showing resistance mostly to tetracycline, oxacillin, erythromycin, clindamycin, kanamycin, gentamicin, ciprofloxacin, chloramphenicol, and sulfamethoxazole trimethoprim. In the study of Nemeghaire et al., only 0.8% of chicken samples were colonized by MRSA in Belgium. However, the strains carried resistance to a wide range of antimicrobial classes and belonged to ST398 and ST239 and spa types t011, t037, and t899 [77]. In another study conducted in chickens in Denmark, the frequency of MRSA detection among chicken samples was 4%. *S. aureus* strains belonged to ST398, ST8, ST5, and ST1, and spa types t011, t108, t008, t347, t002, t034, and t273. Regarding the antimicrobial resistance, the isolates were resistant to tetracycline, erythromycin, clindamycin, and ciprofloxacin [78]. Feßler et al. reported a moderate prevalence (37.2%) of MRSA in chickens and turkeys in Germany. The isolates were ascribed mostly to ST398 but also to ST9, ST5, and ST1791, and *spa* types t011, t034, t899, t023, and t6574. MRSA isolates harboured several virulence genes (*luk*D, *luk*E, *hla*, *hlb*, *can,* and *fib*) and showed resistance to clindamycin, erythromycin, tetracycline, and enrofloxacin [84]. In another study conducted in Germany with chickens and turkeys, several different clonal lineages were detected among the MRSA isolates: ST398, ST5, ST9, ST7, and ST15, and *spa* types t002, t1430, t011, and t034. These MRSA isolates showed resistance to several isolates and a wide diversity of genes encoding virulence factors were also detected, namely, *seg*/i, *sel*m/n/o/u, *luk*F/S/*hlg*A, *luk*D/E, sea, *luk*F/S/*hlg*A, and *sak* [94]. In a more recent study, Silva et al. investigated the prevalence of MRSA in quails. Among the 100 samples tested, 29 were colonized by MRSA. Most strains belonged to ST398 and *spa* type t011 and the remaining isolates were ascribed to ST6831 and *spa* type t9747. All MRSA from quails were multidrug-resistant including resistance to penicillin, ciprofloxacin, erythromycin, clindamycin, tetracycline, aminoglycosides, chloramphenicol, and fusidic acid [88]. In the USA, Waters et al. showed that 54.1% of chickens and turkeys carried MRSA strains which were ascribed to several lineages: ST5, ST6, ST8, ST15, ST406, ST544, ST699, and ST398; however, the spa type was not investigated. The isolates also showed resistance to erythromycin, oxacillin, penicillin, tetracycline, and vancomycin [86]. In 2019, Amoako et al. carried out a study in chickens from South Africa and reported a 20% frequency of MRSA carriage with resistance to cefoxitin, clindamycin, erythromycin, gentamycin, penicillin, and tetracycline. Nevertheless, the genetic lineages were not investigated [83].

As shown in previous studies, MRSA ST2855 is the most prevalent clonal lineage in rabbits. Moreover, MRSA ST2855 has only been reported in studies carried out in wild hares and farm rabbits. Regarding the *spa* type, none showed a significant prevalence. In cattle, MRSA ST398 was the most prevalent, in addition to MRSA ST9 and ST72. MRSA ST398 was initially detected in pigs, however, later it was also detected in companion and food chain animals, as well as in humans, mainly in those who perform professions in which they have frequent contact with animals [95,96]. MRSA ST9 has already been described as a cause of infections in humans and animals, in addition to that it is constantly related to pigs [93,97,98]. Finally, MRSA ST72 is included in the CC8 clonal complex and is found throughout the community and hospitals [99,100]. The most prevalent *spa* types in pigs were t664, t011, and t800. Regarding the studies carried out in pigs, the MRSA that showed a higher prevalence was ST398, which was initially isolated in pigs; later it was isolated in several other animals and even in humans, as previously mentioned. The most prevalent *spa* type was t011, mostly associated with ST398. In Europe, the ST398 clonal lineage is the most frequent lineage detected in MRSA from poultry. Of the 16 studies presented with birds, this lineage was found in 14 (87.5%), followed by ST5 and ST8 (30% and 40%, respectively) [101,102,103]. Resistance among *S. aureus* from birds is very high particularly to penicillin and methicillin; these two antimicrobials are used in most of the studies presented here.

This acquisition of antimicrobial resistance effectively presents a major challenge for the medical world, in human and veterinary terms, regarding the treatment and control of MRSA and *S. aureus* infections.

## Figures and Tables

**Table 1 microorganisms-11-00124-t001:** Mechanisms of action and resistance of the most important classes of antibiotics.

Class of Antibiotics	Mechanisms of Action	Mechanisms of Resistance
Penicillins	Inhibition of cell wall biosynthesis.	Production of β-lactamases; PBP’s changed; efflux pumps.
Cephalosporins		
Aminoglycosides	Inhibition of protein synthesis by binding to the 30S ribosomal subunit.	efflux pumps; modification of the target (ribosome).
Macrolides		
Lincosamides	Inhibition of protein synthesis by binding to the 30S ribosomal subunit.	Modification of the target (ribosome).
Quinolones	Inhibition of DNA synthesis.	Target modification (DNA gyrase and DNA topoisomerase).
Tetracyclines	Inhibition of protein synthesis at the level of peptide elongation.	Efflux pumps; modification of the target (ribosome).
Phenicols	Inhibition of the peptidyltransferase reaction in the 50S ribosomal subunit.	efflux pumps; target modification (enzyme and ribosome).
Sulfonamides	Inhibition of folic acid synthesis.	Target modification (enzymes).

**Table 2 microorganisms-11-00124-t002:** Molecular characteristics of LA-MRSA clones in different continents [22].

Location	Clone	Molecular Characteristics			
		MLST	*spa* Type	SCC*mec* Type	*agr* Type
**USA/** **Canada**	ST398-MRSA V	ST398	t571/t011/t034/t1197/t1250/t1451/t1456/t2510	V	1
	ST5-MRSA IV	ST5	t002/t003/t311	IV	2
**Europe**	ST398-MRSA V	ST398	t571/t011/t034/t1197/t1250/t1451/t1456/t2510	III/IV/V/	1
	ST9	ST9	t100/t411/t899/t4358		2
	ST97-MRSA V	ST97	t1234	V/IV	1
	ST1379-MRSA V	ST1379	t3992	V	1
	ST1-MRSA IV	ST1	t128/t127/t125/t1178	IV	3
	ST130-MRSA XI	ST130	t373	XI	
**Africa**	ST398-MRSA IV	ST398	t571/t011/t034/t1197/t1250/t1451/t1456/t2510	IV	1
**Asia**	ST9-MRSA	ST9	t100/t411/t899/t4358	III/IV/V/NT	2
	ST398-MRSA V	ST398	t571/t011/t034/t1197/t1250/t1451/t1456/t2510	IV	1

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
