# Peer review of "Staphylococcus aureus and MRSA in Livestock: Antimicrobial Resistance and Genetic Lineages"

_microorganisms, 2023, doi:10.3390/microorganisms11010124_

Round 1

Reviewer 1 Report

The manuscript has great relevance and is very well delineated. I request some minor corrections before publication:

Lines 56-60: The abbreviation has already been introduced, there is no need to use the full name.

Line 76: Before using the abbreviation, define what PBP is.

Table 1: Add the corresponding references in each line.

Table 2: Add the corresponding references in each line.

There are two tables named Table 2. Please correct.

Author Response

The manuscript has great relevance and is very well delineated. I request some minor corrections before publication:

A: We appreciate the reviewer’s comments and suggestions.

Lines 56-60: The abbreviation has already been introduced, there is no need to use the full name.

A: Altered in the new version of the manuscript.

Line 76: Before using the abbreviation, define what PBP is.

A: Altered in the new version of the manuscript.

Table 1: Add the corresponding references in each line.

A: Table 1 was adapted from one article which was cited in the table’s caption.

Table 2: Add the corresponding references in each line.

A: Table 2 was also adapted from only one article which was already cited in the table’s caption.

There are two tables named Table 2. Please correct.

A: Altered in the new version of the manuscript.

Reviewer 2 Report

The present manuscript is dedicated on the S. aureus and the methicillin resistant S. aureus in livestock. The authors provide extensive review on the antimicrobial resistance and genetic lineages. Some corrections however should be made:

1. Line 52: Staphylococci an tolerate.... Please, remove "an".

2. Line 222-224: "In this review article we gathered information from studies that detected and characterized S. aureus in pigs, goats, pigs, horses, piglets, sheep, buffaloes, cows, cattle, and rabbits". The animals should be presented in less detail, include poultry as well: "In this review article we gathered information from studies that detected and characterized S. aureus in pigs, goats, horses, sheep, buffaloes,cattle, rabbitts and poultry."Line 227: Rabbits should be with small letter.

Please, correct the number of Tables. There are two tables with number 2. The authors also emphasize much on rabbits, compared to other animals. This should be made in more concise manner.

Author Response

The present manuscript is dedicated on the S. aureus and the methicillin resistant S. aureus in livestock. The authors provide extensive review on the antimicrobial resistance and genetic lineages. Some corrections however should be made:

A: We appreciate the reviewer’s comments and suggestions.

  1. Line 52: Staphylococci an tolerate.... Please, remove "an".

A: Altered in the new version of the manuscript.

  1. Line 222-224: "In this review article we gathered information from studies that detected and characterized S. aureus in pigs, goats, pigs, horses, piglets, sheep, buffaloes, cows, cattle, andrabbits". The animals should be presented in less detail, include poultry as well: "In this review article we gathered information from studies that detected and characterized S. aureus in pigs, goats, horses, sheep, buffaloes,cattle, rabbitts and poultry.

A: Altered in the new version of the manuscript.

"Line 227: Rabbits should be with small letter. 
A: Altered in the new version of the manuscript.

Please, correct the number of Tables. There are two tables with number 2. The authors also emphasize much on rabbits, compared to other animals. This should be made in more concise manner. 

A: Altered in the new version of the manuscript. The emphasis given to rabbits in this review was due to the fact that this is one of the least studied animals.